# Reliability and Validity of the Single-Camera Markerless Motion Capture System for Measuring Shoulder Range of Motion in Healthy Individuals and Patients with Adhesive Capsulitis: A Single-Center Study

**DOI:** 10.3390/s25071960

**Published:** 2025-03-21

**Authors:** Suji Lee, Unhyung Lee, Yohwan Kim, Seungjin Noh, Hungu Lee, Seunghoon Lee

**Affiliations:** 1Department of Acupuncture and Moxibustion Medicine, Kyung Hee University Medical Center, Seoul 02447, Republic of Korea; sjstarry41@naver.com (S.L.); iujhgl@naver.com (U.L.); yoyo4409@naver.com (Y.K.); nohapril@naver.com (S.N.); zxcv219@naver.com (H.L.); 2Department of Acupuncture and Moxibustion, College of Korean Medicine, Kyung Hee University, Seoul 05278, Republic of Korea; 320th Fighter Wing, Republic of Korea Air Force, Seosan 32024, Republic of Korea; 4Department of Clinical Korean Medicine, Graduate School, Kyung Hee University, Seoul 02447, Republic of Korea

**Keywords:** shoulder joint, motion capture, range of motion, reliability, validity

## Abstract

**Highlights:**

**What are the main findings?**
iBalance, a single-camera markerless motion capture system for measuring shoulder range of motion, demonstrated excellent inter- and intra-rater reliability for flexion, abduction, and passive abduction, along with strong validity compared to the goniometer for these movements.Patients with adhesive capsulitis exhibited greater variability than healthy individuals.

**What is the implication of the main finding?**
The iBalance system has the potential to serve as a clinical alternative to goniometers.Its ability to capture dynamic range of motion and compensatory movements offers significant advantages for diagnosing and managing musculoskeletal disorders.

**Abstract:**

Assessing shoulder joint range of motion (ROM) is essential for diagnosing musculoskeletal disorders and optimizing treatments. This single-center pilot study evaluated the reliability and validity of iBalance, a single-camera markerless motion capture system, for measuring shoulder ROM. Forty participants (30 healthy individuals and 10 patients with adhesive capsulitis) underwent measurements of seven shoulder joint movements. Each movement was assessed three times by two raters using both iBalance and a goniometer, with measurements repeated after 1 week. The iBalance demonstrated excellent inter- and intra-rater reliability for flexion (ICC = 0.93 [0.91–0.95], 0.91 [0.88–0.94]), abduction (ICC = 0.97 [0.95–0.98], 0.93 [0.91–0.95]), and passive abduction (ICC = 0.97 [0.96–0.98], 0.98 [0.97–0.98]). The system also showed strong validity compared to the goniometer for flexion (ICC = 0.85 [0.68–0.92]), abduction (ICC = 0.95 [0.94–0.96]), and passive abduction (ICC = 0.97 [0.96–0.98]). Bland–Altman plots showed high consistency between the two devices for flexion, abduction, and passive abduction, with most data points falling within the limits of agreement. Patients with adhesive capsulitis exhibited greater variability than healthy individuals. No adverse events were reported, supporting the safety of the system. This study highlights the potential of a single-camera markerless motion capture system for diagnosing and treating shoulder joint disorders. The iBalance showed clinical applicability for measuring flexion, abduction, and passive abduction. Future enhancements to the algorithm and the incorporation of advanced metrics could improve its performance, facilitating broader clinical applications for diagnosing complex shoulder conditions.

## 1. Introduction

Assessment of shoulder joint range of motion (ROM) is essential for diagnosing musculoskeletal disorders, supporting injury rehabilitation, and optimizing exercise performance [1]. Traditionally, ROM has been measured using goniometers; however, this method has notable limitations, such as inter-rater variability, time-consuming procedures, and challenges in stabilizing the trunk and shoulder blades, which can lead to measurement errors. Additionally, shoulder movement involves a complex interaction among multiple joints, including the clavicle, scapula, and humerus, collectively referred to as the scapulohumeral rhythm. Thus, assessing shoulder movement in a two-dimensional manner using a goniometer may not be appropriate [2]. To address these limitations, alternative ROM assessment tools such as inertial sensors and digital inclinometers have been introduced. Inertial sensors, which integrate accelerometers and gyroscopes, provide a portable and cost-effective solution for motion analysis, particularly in shoulder disorders such as adhesive capsulitis [3]. Similarly, digital inclinometers offer high accuracy in measuring scapular movement and are widely used in rehabilitation settings [4]. However, both methods require direct sensor placement on the body, which can lead to variability in measurements due to sensor positioning and movement artifacts.

In response to these issues, marker-based motion capture systems with reflective skin markers have been developed to significantly improve measurement accuracy. However, these systems face challenges, such as the inconvenience of marker attachment, interference with participant movement, and the need for ample space and a controlled environment. For these reasons, their application in clinical settings remains challenging and often impractical [5]. Moreover, most motion analysis studies have primarily focused on assessing ROM in healthy individuals, with limited research on patients with shoulder disorders. Adhesive capsulitis, or frozen shoulder, is a common musculoskeletal condition characterized by progressive pain and restricted active and passive ROM without structural lesions [6]. Since passive ROM assessment is crucial for evaluating disease progression and treatment response, expanding motion analysis research to include both active and passive ROM in adhesive capsulitis patients is essential.

Recently, markerless motion capture techniques for ROM assessment have gained considerable attention as alternatives to traditional systems. Depth sensor-based motion analysis systems utilizing RGB-D cameras, such as Kinect (Microsoft Corp., Redmond, WA, USA) and RealSense (Intel Corp., Santa Clara, CA, USA), have emerged as promising tools. These cameras combine an RGB sensor for capturing color images with a depth sensor employing structured light or time-of-flight technology, generating detailed three-dimensional (3D) representations of the environment. By processing these data, RGB-D systems can accurately identify joint positions and measure kinematics, offering non-invasive, efficient, and practical solutions for both clinical and research applications [7,8].

Although many studies [9,10,11,12] have evaluated the reliability and validity of shoulder joint ROM measurement using RGB-D cameras, their findings are inconsistent. Some report moderate to excellent intraclass correlation coefficients (ICCs) for movements like flexion and abduction [10,12], while others highlight significant measurement errors due to depth estimation inaccuracies and joint occlusions [9,11]. A systematic review and meta-analysis [13] further underscored these inconsistencies, reporting that inter-rater reliability for the Kinect ranged from moderate to good (e.g., for flexion and abduction), whereas intra-rater reliability varied significantly, ranging from poor (ICC < 0.65) to good (ICC > 0.85). Additionally, the lack of standardized protocols and inconsistent methodologies contributed to high heterogeneity among the findings. These limitations emphasize the need for well-designed studies to establish more consistent and clinically applicable outcomes for RGB-D camera assessments.

iBalance is an advanced program designed for accurate 3D human pose estimation using a single camera, utilizing deep learning techniques and RGB-D data from a Kalman filter-based multiple-camera system. This program recognizes joints, establishes their axes, and measures angles, enabling rapid ROM and posture assessments without the need for markers. By integrating advanced pose-estimation algorithms, a two-dimensional (2D) to 3D lift-up algorithm, and a self-developed 3D transformation model, iBalance achieves precise joint coordinate calculation with minimal computational load. Additionally, techniques such as median filtering and exponential moving averages enhance stability in posture estimation under challenging conditions. Compared to existing RGB-D-based systems, iBalance provides improved reliability and precision, establishing itself as a more practical tool for real-time motion analysis in clinical and research settings.

Thus, the primary aim of this study was to evaluate the reliability and validity of iBalance for measuring shoulder joint ROM in clinical settings. Additionally, we explored whether the system’s measurement reliability and validity showed different trends between healthy individuals and patients with adhesive capsulitis.

## 2. Material and Methods

### 2.1. Study Design

This single-center pilot study assessed the reliability and validity of a single-camera markerless motion capture system. Participants were recruited via posters and advertisements at Kyung Hee University Korean Medicine Hospital. Researchers measured shoulder joint ROM using both the single-camera markerless motion capture system and a goniometer. The complete schedule of the study participants is presented in Figure 1.

### 2.2. Ethics Approval and Consent to Participate

The study protocol was approved by the Institutional Review Board of Kyung Hee University Korean Medicine Hospital (KOMCIRB 2022-06-003) on 22 July 2022. It was registered in the Korean Clinical Trials Registry (KCT0008251). Participants received detailed information about the study, and written informed consent was obtained before enrollment.

### 2.3. Participants

This study included 40 participants aged 19 years or older, consisting of 30 healthy individuals and 10 patients with adhesive capsulitis. Healthy individuals were eligible if they had no history of shoulder joint disease or pain. Patients with adhesive capsulitis in the freezing or frozen stage were included based on the presence of passive motion restrictions in at least two directions (flexion < 165°, abduction < 150°, external rotation < 45°) lasting more than 1 month [14]. Participants were excluded if they had a history of major trauma involving the shoulder, symptomatic rotator cuff tears confirmed by ultrasound or MRI, inflammatory joint diseases, or any other structural or systemic disorder that could contribute to shoulder pain or restricted ROM. Additionally, individuals from both groups were excluded if they had difficulty complying with the study schedule or if they had severe musculoskeletal pain, neurological disorders affecting motor function, cognitive impairments interfering with test instructions, or other medical conditions that could compromise accurate ROM assessment.

### 2.4. Measurement Systems

iBalance (version 1.1.0), a single-camera markerless motion capture system, is medical software developed by TeamElysium Inc. (Seoul, Republic of Korea) that specializes in postural measurements. The system captures high-resolution RGB images (1280 × 720, 38 bit) and depth images (1280 × 720, 116 bit), enabling detailed joint movement analysis. The 2D detection process extracts key points from 32 major body joints, including the pelvis, spine, shoulders, elbows, wrists, and ankles. After extracting the 2D key points, the 3D transformation step inputs these points along with nearby depth values into a proprietary 3D transformation model to estimate 3D coordinates with minimal computation. To ensure robust posture estimation, the software employs a median filter, exponential moving average, and test-time augmentation. Compared to existing software, iBalance enhances the stability of 2D key point extraction and provides highly reliable pose estimation while maintaining simplicity and accuracy in the 3D transformation process. This ensures stable pose analysis results, even in real-time motion analysis environments or scenarios with limited training data.

Since this is the first study utilizing the iBalance system, we referenced studies using similar markerless RGB depth sensor-based motion capture systems, which have already shown high validity and reliability in measuring shoulder kinematics [15,16]. The camera was positioned 95 cm above the ground and aligned perpendicularly, with a distance of 250 cm from the participant to capture their entire body (Figure 2). Measurement accuracy was influenced by environments with poor infrared visibility or by unsuitable clothing, such as black fabrics. To minimize these issues, participants were advised to avoid reflective or infrared-absorbing materials per the manufacturer’s guidelines. Standardized clothing was not required, as the study aimed to reflect real-world clinical conditions where patients do not typically change attire. Test clothing was provided when needed.

iBalance was compared with a 12-inch plastic goniometer (Model 12-1000, Base-line, Garden City, NY, USA), which served as the standard reference for ROM measurement.

### 2.5. Procedures

Two Korean Medicine doctors with 2 years of experience conducted the study after sufficient practice based on measurement protocol. The raters were blinded to the results, and a third independent researcher recorded all data. Seven shoulder joint movements (active flexion, extension, abduction, adduction, external rotation, internal rotation, and passive abduction) were measured using both iBalance and a goniometer (Figure 3).

Each movement was measured three times per session to ensure consistency and stability within a single session. Detailed information on the motion measurements is presented in the Appendix A, with additional analyses of the repeated measurements provided in Appendix A. For healthy participants, both shoulders were measured, while for patients with adhesive capsulitis, only the affected shoulder was assessed. Both raters performed the entire procedure identically. A third researcher randomized both the raters and the measurement order using a random table. Participants were allowed a 3 min rest if they experienced fatigue or pain during the measurements. Symptoms such as pain or joint sounds were monitored during each session. All measurements were repeated after 7 days. This process is illustrated in Figure 4.

### 2.6. Outcomes

The outcomes of the study were the angle values obtained from three repeated measurements of shoulder joint ROM. Absolute reliability was assessed using the mean absolute deviation (MAD) with a 10% margin for acceptance, the standard error of the mean (SEM), minimum detectable change (MDC), and limit of agreement (LOA). Relative reliability, including test–retest, intra-rater, and inter-rater reliability, was assessed using the ICC. The validity was determined by comparing the results of iBalance with those of the goniometer using ICC, Bland–Altman plots, and LOA.

### 2.7. Statistical Analysis

Block randomization was applied to prevent memory effects that might influence measurement reliability. For missing values or participant dropouts, the mean of the remaining measurements was used to impute missing data. Outliers were excluded based on predefined criteria [17]. Common outlier criteria included flexion values of 0° or > 180°, extension values of 0° or ≥60°, abduction values of 0° or >180°, adduction values of 0° or ≥60°, external rotation values of 0° or ≥90°, and internal rotation values of 0° or ≥90°. Additional criteria for healthy participants included abnormally low ROM values, defined as flexion <60°, extension ≤20°, abduction <60°, adduction ≤20°, external rotation ≤30°, and internal rotation ≤30°. Absolute reliability was evaluated using the MAD, SEM, MDC, and LOA. The equations were as follows:$$MAD=\frac{1}{N}\sum_{i=1}^{N}\left|X_{i}-Mean\;value,\right|$$
where N: total number of data values, X_i_: each data value.$$SEM=SD\times \sqrt{1-ICC,}$$$$MDC=1.96\times SEM\times \sqrt{2,}$$LOA=Mean difference±1.96×SD.

Relative reliability, including test–retest, intra-rater, and inter-rater reliability, was assessed using ICC (2,1) based on a two-way random single-measures model. Validity was evaluated by comparing the results of iBalance with those of the goniometer using ICC, Bland–Altman plots, and LOA. The ICC values were interpreted for reliability and validity using the following criteria: excellent (0.75–1.00), good (0.60–0.75), fair (0.40–0.60), and poor (<0.40) [18]. Statistical analyses were performed using SAS (version 9.4; SAS Institute Inc., Cary, NC, USA) or R (version 4.0.0).

### 2.8. Sample Size

The sample size calculation was based on reliability and validity analyses. According to Landis and Koch [19], a reliability coefficient above the moderate level was considered reliable. To achieve 80% power to detect an ICC of 0.75, assuming a null hypothesis ICC of 0.50, a sample size of 36 participants with two raters was determined to be sufficient based on an F-test with a significance level of 0.05. Considering the 10% dropout rate, the final sample size was set at 40. Since this study aimed to evaluate the reliability and validity of the system rather than to compare groups statistically, the sample size was determined based on achieving sufficient power for reliability analysis. A 3:1 ratio of healthy individuals (n = 30) to patients with adhesive capsulitis (n = 10) was chosen to ensure the inclusion of individuals with restricted ROM, allowing for a broader evaluation of the system’s performance across different movement conditions.

## 3. Results

Forty volunteers (30 healthy participants and 10 patients with adhesive capsulitis) were included in this study. There were no significant differences between the groups in variables, except for age, reflecting the tendency of adhesive capsulitis to occur more frequently in middle-aged individuals (Table 1).

### 3.1. Reliability

Using the iBalance system to measure shoulder ROM, all movements demonstrated excellent test–retest reliability. Both inter-rater and intra-rater reliability were excellent for flexion, abduction, and passive abduction. For extension and external rotation, inter-rater reliability remained excellent, but intra-rater reliability was classified as fair. For adduction, both inter-rater and intra-rater reliability were classified as good, while for internal rotation, inter-rater reliability was good, and intra-rater reliability was fair. Absolute reliability was assessed using MAD with a 10% margin for acceptance, as well as SEM and MDC. For flexion, abduction, and passive abduction, MAD values were relatively low, and SEM and MDC values were within acceptable ranges, indicating high measurement precision and consistency. In contrast, extension and internal rotation exhibited higher MAD and SEM values, suggesting greater variability and lower precision for these movements (Table 2).

Measurements obtained using the goniometer showed similar reliability patterns to those observed with the iBalance (Appendix A). Specifically, inter-rater reliability for extension and internal rotation was classified as good, while intra-rater reliability was fair. For adduction, only intra-rater reliability was classified as fair. Both methods demonstrated relatively high reliability for flexion, abduction, and passive abduction. The analysis was further stratified by hand dominance, including the dominant hand, right hand, and left hand (Appendix A). The results were consistent across the movements, showing no significant differences based on hand dominance. Among patients with adhesive capsulitis, inter-rater reliability was excellent for most movements, except for adduction and internal rotation. However, intra-rater reliability was generally lower, with only flexion, abduction, and passive abduction showing acceptable levels (Appendix A).

### 3.2. Validity

When assessing the validity of iBalance against the goniometer as a reference using ICC criteria, flexion, abduction, external rotation, and passive abduction demonstrated excellent validity, while adduction showed fair validity. In contrast, extension and internal rotation were classified as poor (Table 3). The analysis was further subdivided by hand dominance, which included the dominant hand, right hand, and left hand (Appendix A). The findings were consistent across the movements, showing no significant differences based on hand dominance. Among patients with adhesive capsulitis, excellent validity was observed only for abduction and passive abduction movements (Appendix A).

In the Bland–Altman plot, most data for flexion, abduction, and passive abduction fell within the LOA range, indicating high consistency between the two devices. However, external rotation exhibited relatively larger measurement discrepancies. Overall, patients with adhesive capsulitis showed greater variability compared to healthy individuals (Figure 5, Appendix A).

### 3.3. Adverse Events

During the study, no adverse events, such as pain complaints or worsening of symptoms while repeating the movements, were reported.

## 4. Discussion

Accurate measurement of shoulder joint ROM is crucial for diagnosing and managing musculoskeletal conditions. In particular, adhesive capsulitis is characterized by progressive stiffness and pain, making ROM assessment essential for diagnosis and treatment monitoring. Including patients with adhesive capsulitis allowed us to observe how the system performs under conditions of restricted mobility, providing additional insights into its clinical applicability.

This study evaluated the reliability and validity of iBalance, a markerless motion capture system, for assessing shoulder ROM compared to a goniometer. iBalance demonstrated excellent inter- and intra-rater reliability for flexion, abduction, and passive abduction, with acceptable reliability for extension, adduction, and internal rotation. This consistent performance across multiple movements highlights the system’s potential to provide precise and reproducible measurements in clinical settings. In addition to reliability, iBalance exhibited excellent validity compared to the goniometer for flexion, abduction, external rotation, and passive abduction. However, the LOA values revealed that iBalance consistently measured higher ROM values than the goniometer. This discrepancy reflects differences in measurement principles, where iBalance captures peak ROM during dynamic motion, whereas goniometers rely on static measurements at specific positions. Similarly, a previous study using Kinect technology reported that markerless motion capture systems produced results more comparable to active ROM than passive ROM, supporting the dynamic measurement capabilities of such systems [14]. While this suggests that iBalance may provide a more realistic representation of dynamic joint function, further standardization in interpreting these differences is necessary.

To achieve these results, this study implemented a robust protocol designed to minimize bias and enhance reliability. Two independent raters performed three measurements for all shoulder movements at 7-day intervals, with randomized rater order. Absolute reliability metrics, including MAD, a 10% margin, SEM, and MDC, were also incorporated to comprehensively evaluate the system’s consistency. Compared to previous studies, which often report inconsistent inter-rater reliability for goniometers [13], this approach demonstrates the methodological advantages of iBalance in producing stable and reproducible results.

In patients with adhesive capsulitis, iBalance exhibited good reliability for flexion, abduction, and passive abduction, though overall reliability and validity scores were lower than those for healthy participants. Validity was excellent only for abduction and passive abduction, while other movements demonstrated weaker validity. These findings are consistent with a previous study [14] that reported high variability in ROM measurements among patients with adhesive capsulitis, likely due to differences in disease progression, stages, and joint stiffness. Additionally, pain severity and fluctuation may contribute to this variability, particularly in intra-rater reliability. Tveita et al. [20] found that while ROM measurements in adhesive capsulitis were reproducible at the group level, individual variability was substantial. This suggests that clinicians should be cautious when assessing ROM in individual patients, as disease progression and pain responses may lead to significant measurement fluctuations. Furthermore, a study utilizing depth sensor-based motion analysis [21] has shown that patients with adhesive capsulitis exhibit significantly lower maximum abduction angles and angular velocities compared to both healthy individuals and other shoulder pathology groups. These findings indicate that the observed reductions in ROM and movement speed among patients with adhesive capsulitis are primarily driven by pathological changes in joint mechanics rather than inconsistencies in measurement methodology. This highlights the potential of markerless motion capture systems like iBalance to differentiate movement abnormalities across various shoulder disorders.

Recent advancements in 3D depth sensor technology, such as iBalance, have expanded the possibilities for motion analysis beyond traditional ROM measurements. These systems allow for dynamic motion tracking and provide insights into metrics such as angular velocity and time-to-motion ratio. For example, a prior study [21] demonstrated that patients with adhesive capsulitis exhibit slower angular velocities and prolonged time-to-motion ratios during abduction and adduction, suggesting that these metrics could enhance diagnostic accuracy. Additionally, 3D systems enable the assessment of inter-joint coordination, such as scapulothoracic contributions during shoulder movements or compensatory patterns like trunk involvement during arm elevation. These advanced capabilities offer clinicians a deeper understanding of movement mechanics and facilitate more tailored treatment approaches.

Despite these strengths, this study has certain limitations. First, the single-camera design of iBalance presents challenges for movements like extension and internal rotation, which are prone to occlusion errors [22]. Incorporating multi-camera systems or refining algorithms to improve tracking of small joint angles could help address these limitations. Second, the relatively small sample size, especially for patients with adhesive capsulitis, limits the generalizability of the findings. While the 3:1 ratio was chosen based on recruitment feasibility, a more balanced sample may provide stronger statistical comparisons. Third, although this study included both healthy individuals and patients with adhesive capsulitis, differences in age and gender were not separately analyzed. Descriptive statistics showed that age differences existed between groups, which may have influenced ROM variability. Future studies should consider age and gender differences to better understand their potential impact on ROM measurements. Finally, comparisons with marker-based motion tracking systems, considered the gold standard for dynamic movement analysis, could provide a more rigorous benchmark for validation.

## 5. Conclusions

iBalance demonstrated excellent reliability and validity for flexion, abduction, and passive abduction, establishing its potential as a clinical alternative to goniometers. Its ability to capture dynamic ROM and compensatory movements offers significant advantages for diagnosing and managing musculoskeletal disorders. Future studies should focus on optimizing algorithms, expanding sample sizes, and exploring the applicability of iBalance across diverse clinical conditions. With continued advancements, iBalance holds promise for transforming musculoskeletal assessments and improving patient outcomes.

## Figures and Tables

**Figure 1 sensors-25-01960-f001:**
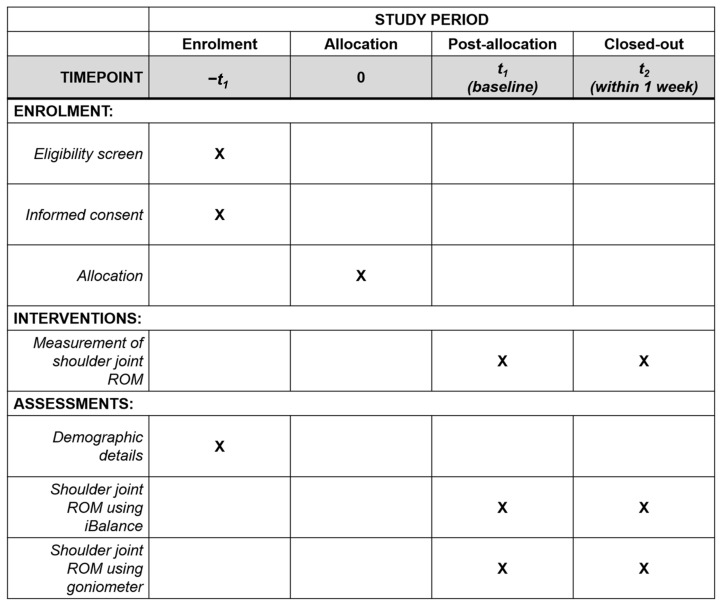
Timeline and protocol for shoulder ROM measurement.

**Figure 2 sensors-25-01960-f002:**
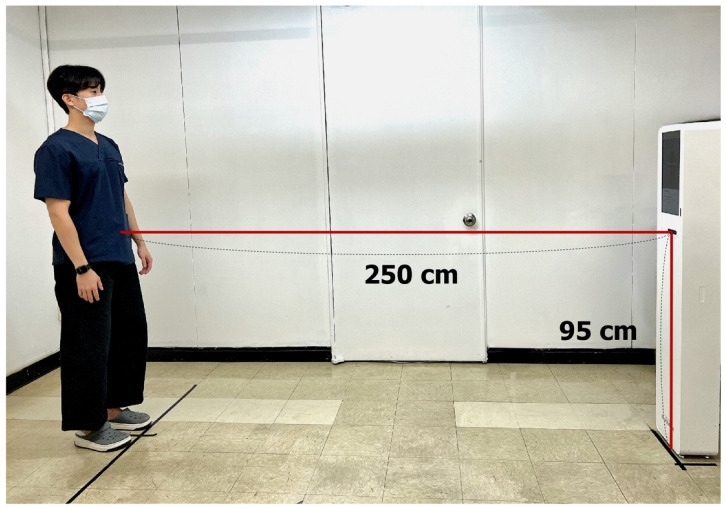
iBalance setting.

**Figure 3 sensors-25-01960-f003:**
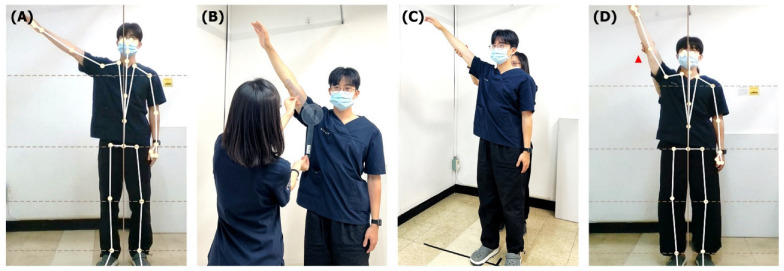
Measurement of shoulder range of motion. (**A**) Active abduction is measured using the iBalance. (**B**) Active abduction is measured using a goniometer. (**C**) The posture of the participant and assistant during passive abduction. (**D**) Passive abduction is measured using the iBalance. The movement will be performed with the help of an assistant, and it will be confirmed that the assistant’s hand, indicated by the red arrowhead, is not recognized by the device.

**Figure 4 sensors-25-01960-f004:**
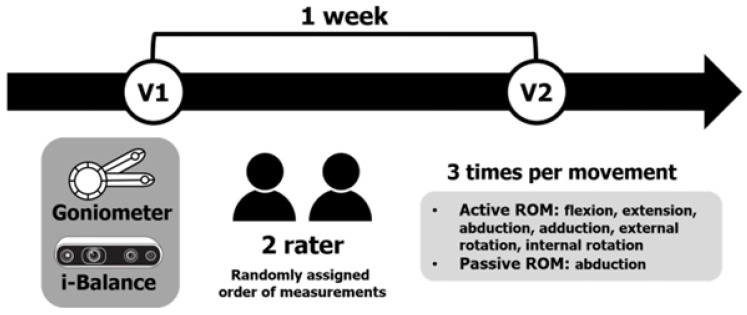
Test procedure. V, visit; ROM, range of motion.

**Figure 5 sensors-25-01960-f005:**
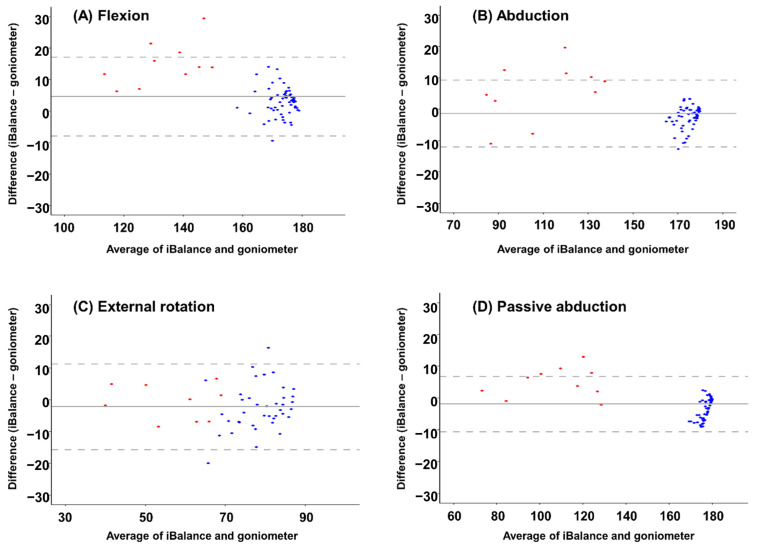
(**A**) Flexion, (**B**) Abduction, (**C**) External rotation, (**D**) Passive abduction. Bland–Altman plots illustrate the validity of shoulder range of motion measurements. Blue dots represent healthy individuals, while red dots represent patients with adhesive capsulitis. The solid line indicates the mean difference between the two devices, and the dashed lines represent the limits of agreement (mean ± 1.96 × standard deviation).

**Table 1 sensors-25-01960-t001:** Baseline Demographic and Clinical Characteristics of Healthy Individuals and Patients with Adhesive Capsulitis.

Variable	Healthy (n = 30)	Patients with Adhesive Capsulitis (n = 10)	*p*-Value
Sex (Male/Female)	6/24	2/8	1.00
Dominant hand (Left/Right)	2/28	1/9	0.73
Age (mean ± SD)	30.87 ± 5.99	53.60 ± 6.55	<0.0001 *
Height (mean ± SD)	165.24 ± 7.77	160.62 ± 7.88	0.11
Weight (mean ± SD)	60.45 ± 14.07	57.27 ± 10.25	0.65

*p*-values < 0.05 are marked with an asterisk (*). SD, standard deviation.

**Table 2 sensors-25-01960-t002:** Reliability Results of iBalance for Shoulder Range of Motion Measurements.

		ICC [2, 1] (95% CI)	MAD	Margin 10%	*p*-Value	SEM	MDC
Flexion	Inter-rater	0.93 (0.91–0.95)	3.40	17.00	<0.0001 *	3.67	10.17
	Intra-rater	0.91 (0.88–0.94)	4.13	17.01	<0.0001 *	4.20	11.65
Extension	Inter-rater	0.77 (0.69–0.83)	4.55	3.62	0.9958	4.26	11.81
	Intra-rater	0.54 (0.40–0.65)	7.09	3.62	1.0000	6.17	17.11
Abduction	Inter-rater	0.97 (0.95–0.98)	3.15	16.55	<0.0001 *	4.28	11.86
	Intra-rater	0.93 (0.91–0.95)	4.61	16.60	<0.0001 *	5.75	15.95
Adduction	Inter-rater	0.70 (0.60–0.78)	4.71	3.03	1.0000	4.29	11.89
	Intra-rater	0.64 (0.52–0.73)	5.28	3.06	1.0000	4.80	13.32
External rotation	Inter-rater	0.78 (0.65–0.86)	6.59	6.89	0.3683	6.53	18.10
	Intra-rater	0.58 (0.33–0.75)	10.36	6.70	0.9967	9.17	25.41
Internal rotation	Inter-rater	0.73 (0.63–0.81)	4.36	7.26	<0.0001 *	4.09	11.35
	Intra-rater	0.56 (0.42–0.67)	5.54	7.25	0.0001 *	5.14	14.25
Passive abduction	Inter-rater	0.97 (0.96–0.98)	3.51	16.68	<0.0001 *	4.05	11.23
	Intra-rater	0.98 (0.97–0.98)	2.93	16.73	<0.0001 *	3.49	9.68

*p*-values < 0.05 are marked with an asterisk (*). The ICC values were classified for reliability and validity based on the following criteria: excellent (0.75–1.00), good (0.60–0.75), fair (0.40–0.60), and poor (<0.40). ICC, intraclass correlation coefficient; CI, confidence interval; MAD, mean absolute deviation; SEM, standard error of the mean; MDC, minimum detectable change.

**Table 3 sensors-25-01960-t003:** Validity of iBalance for shoulder range of motion measurements compared to goniometer.

	ICC [2, 1] (95% CI)	LOA	
		Lower	Upper
Flexion	0.85 (0.68–0.92)	−5.17	18.01
Extension	0.25 (−0.05–0.48)	−3.15	23.80
Abduction	0.95 (0.94–0.96)	−5.85	16.02
Adduction	0.42 (−0.05–0.68)	−2.35	19.22
External rotation	0.75 (0.65–0.82)	−4.82	19.72
Internal rotation	0.23 (−0.08–0.48)	−2.80	22.89
Passive abduction	0.97 (0.96–0.98)	−4.34	12.44

The ICC values were classified for reliability and validity based on the following criteria: excellent (0.75–1.00), good (0.60–0.75), fair (0.40–0.60), and poor (<0.40). ICC, intraclass correlation coefficient; CI, confidence interval; LOA, limit of agreement.

## Data Availability

The data presented in this study are available upon request from the corresponding author.

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
