# Peer review of "Reliability and Validity of the Single-Camera Markerless Motion Capture System for Measuring Shoulder Range of Motion in Healthy Individuals and Patients with Adhesive Capsulitis: A Single-Center Study"

_sensors, 2025, doi:10.3390/s25071960_

Round 1
Reviewer 1 Report
Comments and Suggestions for Authors
In the presented paper, the authors have conducted a study to evaluate the reliability and validity of using a single-camera markerless motion capture system to measure the range of motion of the shoulder joint.
The study is novel and the need is well-justified in the introduction section. Although the results are clearly presented and the conclusion reached is justified, there are some concerns regarding the study itself. For instance, the ratio of control group to patients is 3:1, it would be better if the sample size for the affected participants is closer to that of the control group. It would make for better comparison and increase the validity of the results. It should be noted that the authors have also identified this limitation and it is hoped that changes will be made for future studies.
Also, during the data collection stage, it is mentioned that wearing appropriate clothing was optional and only based on need. It would have been best if all participants were instructed to wear the same clothing thereby making it a more standard study which could enhance the validity and repeatability of the results.
There are also minor spelling errors which should be corrected during editing.
Overall, this was a a well-conducted study and shows that the Shoulder range of motion measurements obtained from a single-camera markerless motion capture system has good validity and reliability.
Author Response
Thank you for your valuable feedback on our manuscript. We appreciate your constructive comments, which have helped us refine our study and improve its clarity. Below, we respond to your concerns and describe the revisions made accordingly.
- Control-to-Patient Ratio
Reviewer’s Comment:
"The ratio of the control group to patients is 3:1. It would be better if the sample size for the affected participants is closer to that of the control group. It would make for better comparison and increase the validity of the results."
Response:
We acknowledge the discrepancy in sample sizes between the control and patient groups. However, as this study is a preliminary investigation, our primary objective was to evaluate the reliability and validity of the single-camera markerless motion capture system, rather than to conduct direct statistical comparisons between normal individuals and patients with frozen shoulder. The patient group was included to explore whether the system’s reliability and validity might be influenced by restricted range of motion, rather than to compare the two groups statistically.
Additionally, our focus was to observe potential trends in system performance under different conditions, rather than to establish statistical differences. We have clarified this point in the manuscript for better understanding.
Revision in the manuscript (2.8. Sample Size)
Original:
“This study included both healthy individuals (n = 30) and patients with adhesive capsulitis (n = 10), with a 3:1 ratio chosen based on recruitment feasibility to address the lack of data on patient populations in previous studies.”
Updated:
“Since this study aimed to evaluate the reliability and validity of the system rather than to compare groups statistically, the sample size was determined based on achieving sufficient power for reliability analysis. A 3:1 ratio of healthy individuals (n = 30) to patients with adhesive capsulitis (n = 10) was chosen to ensure the inclusion of individuals with restricted ROM, allowing for a broader evaluation of the system's performance across different movement conditions.”
- Standardization of Participant Clothing
Reviewer’s Comment:
"During the data collection stage, it is mentioned that wearing appropriate clothing was optional and only based on need. It would have been best if all participants were instructed to wear the same clothing, thereby making it a more standard study which could enhance the validity and repeatability of the results."
Response:
We appreciate this suggestion and recognize the benefits of standardized participant clothing in experimental settings. However, in this study, we aimed to replicate real-world clinical conditions, where patients typically do not change into standardized clothing before motion analysis. One of the key advantages of this markerless motion capture system is its ability to be used in routine clinical practice without requiring specialized attire.
To minimize potential variability, we followed the manufacturer’s recommendations on appropriate clothing and excluded participants whose attire might interfere with motion tracking (e.g., loose-fitting garments obscuring limb contours). We have now explicitly mentioned this in the manuscript.
Revision in the manuscript (2.4. Measurement systems)
Original:
“Measurement accuracy was influenced by environments with poor infrared visibility or by unsuitable clothing, such as black fabrics. Participants were instructed to avoid reflective or infrared-absorbing materials to minimize these issues, and test clothing was provided when necessary.”
Updated:
“Measurement accuracy was influenced by environments with poor infrared visibility or by unsuitable clothing, such as black fabrics. To minimize these issues, participants were advised to avoid reflective or infrared-absorbing materials per the manufacturer’s guidelines. Standardized clothing was not required, as the study aimed to reflect real-world clinical conditions in which patients typically undergo motion analysis without changing clothes. Test clothing was provided when needed.”
- Minor Spelling Errors
Reviewer’s Comment:
"There are also minor spelling errors which should be corrected during editing."
Response:
Thank you for bringing this to our attention. We have carefully reviewed the manuscript and corrected all spelling and grammatical errors to enhance readability and clarity.
We sincerely appreciate your insightful comments, which have helped us clarify the study’s objectives and methodology. We have revised the manuscript to address your concerns and improve transparency. We hope our responses satisfactorily address your feedback and that the revised manuscript meets your expectations.
Reviewer 2 Report
Comments and Suggestions for Authors
Dear authors,
First of all thank you for the invitation to review your study “ Reliability and Validity of the Single-camera Markerless Motion Capture System for Measuring Shoulder Range of Motion: A Single-center Study”. The following suggestions could improve the quality of your work.
TITLE: the study involved subject with adhesive capsulitis, I think that this should be added in your title
ABSTRACT
- I would add more statistical results in order to make your abstract more attractive.
INTRODUCTION
- Again, the study involved subject with adhesive capsulitis, that represents the most common musculoskeletal condition of shoulder pain and dysfunction, especially for range of motion. In fact, this condition is also called “frozen shoulder” due to the important reduction in the range of motion. Therefore, ROM represent with pain the most important outcomes to consider in the treatment of adhesive capsulitis. I would add a paragraph of this condition in order to support your study.
- “Traditionally, ROM has been measured using goniometers” ok, but there are others method to assess ROM, for example Inertial sensor that combine easy accessibility due to low cost and ease of transport and use while performing motion analysis with higher reliability. An other possibility is the digital inclinometers that are demonstrated to be a reliable tool for the measurement of scapular upward rotation. Please add this two tools and thake in to consideration the following articles that use these tools in adhesive capsulitis subjects:
Deodato M, Martini M, Buoite Stella A, Citroni G, Ajčević M, Accardo A, Murena L. Inertial Sensors and Pressure Pain Threshold to Evaluate People with Primary Adhesive Capsulitis: Comparison with Healthy Controls and Effects of a Physiotherapy Protocol. J Funct Morphol Kinesiol. 2023 Oct 6;8(4):142. doi: 10.3390/jfmk8040142. PMID: 37873901; PMCID: PMC10594492.
Mohamed AA, Jan YK, El Sayed WH, Wanis MEA, Yamany AA. Dynamic scapular recognition exercise improves scapular upward rotation and shoulder pain and disability in patients with adhesive capsulitis: a randomized controlled trial. J Man Manip Ther. 2020 Jul;28(3):146-158. doi: 10.1080/10669817.2019.1622896. Epub 2019 Jun 14. Erratum in: J Man Manip Ther. 2020 Jul;28(3):159. doi: 10.1080/10669817.2020.1764269. PMID: 31200629; PMCID: PMC7480516.
- Aim, I think that your aim is to compare the ROM with goniometer with respect to Markerless Motion Capture System in the shoulders of patients with adhesive capsulitis and in the shoulders of healthy controls. Please describe better your aim. You can split in a primary and secondary aim.
METHOD
- “Researchers measured shoulder joint ROM” could you kindly specify if researcher are physiotherapist, engineers or others?
- The manuscript does not clearly justify the chosen sample size or discuss how it ensures sufficient power to detect expected differences between groups.
- inclusion/exclusion criteria should be better explain
- please specify in which phase are the patients with adhesive capsulitis
- 2.4. Measurement Systems: please add more references that support the use of this Markerless Motion Capture System
RESULT
- The quality of your tables should be improved, please use only three lines
DISCUSSION
- I would add a discussion concerning the importance to assess the ROM in Adhesive capsulitis.
- Your discussion would benefit from a comparison with others study involved in other assessment of ROM in adhesive capsulitis and in healthy controls, you can use also the studies suggested in the introduction section.
- The main limitation of your study is the sample size that does not allow age, gender stratification nor a comparison between healthy and adhesive capsulitis.
Author Response
Thank you for your valuable feedback on our manuscript. We appreciate your constructive comments, which have helped us refine our study and improve its clarity. Below, we provide responses to your concerns and describe the revisions made accordingly.
- Title Revision
Reviewer’s Comment:
"The study involved subjects with adhesive capsulitis. I think that this should be added in your title."
Response:
We acknowledge that our study includes both healthy individuals and patients with adhesive capsulitis. To better reflect the study population, we have revised the title accordingly.
Revision in the manuscript:
Original:
"Reliability and Validity of the Single-Camera Markerless Motion Capture System for Measuring Shoulder Range of Motion: A Single-Center Study"
Updated:
"Reliability and Validity of the Single-Camera Markerless Motion Capture System for Measuring Shoulder Range of Motion in Healthy Individuals and Patients with Adhesive Capsulitis: A Single-Center Study"
This revision explicitly includes both healthy individuals and patients with adhesive capsulitis while maintaining clarity and conciseness.
- Abstract Revision
Reviewer’s Comment:
"I would add more statistical results in order to make your abstract more attractive."
Response:
We appreciate your suggestion and have revised the abstract by incorporating key statistical results to enhance clarity and informativeness. Specifically, we have added intraclass correlation coefficient (ICC) values to highlight reliability and validity results and included 95% limits of agreement ranges to better illustrate measurement consistency.
Revision in the manuscript:
Original:
iBalance demonstrated excellent inter- and intra-rater reliability for flexion, abduction, and passive abduction (ICC>0.90) and strong validity compared to the goniometer for the same movements (ICC>0.85). Bland–Altman plots indicated high consistency between the two devices for these movements, with most data points falling within the limits of agreement.
Updated:
iBalance demonstrated excellent inter- and intra-rater reliability for flexion (ICC = 0.93 [0.91-0.95], 0.91 [0.88-0.94]), abduction (ICC = 0.97 [0.95-0.98], 0.93 [0.91-0.95]), and passive abduction (ICC = 0.97 [0.96-0.98], 0.98 [0.97-0.98]). The system also showed strong validity compared to the goniometer for flexion (ICC = 0.85 [0.68-0.92]), abduction (ICC = 0.95 [0.94-0.96]), and passive abduction (ICC = 0.97 [0.96-0.98]).
- Introduction Revision
Reviewer’s Comments:
- "The study involved subjects with adhesive capsulitis, which is one of the most common musculoskeletal conditions associated with shoulder pain and dysfunction. ROM measurement is critical in its diagnosis and treatment evaluation. Please add a paragraph explaining this condition to support your study."
- "Traditionally, ROM has been measured using goniometers, but other tools exist, such as inertial sensors and digital inclinometers, which offer advantages in reliability and ease of use. Please mention these tools and consider citing relevant studies."
- "Aim, I think that your aim is to compare the ROM with goniometer with respect to Markerless Motion Capture System in the shoulders of patients with adhesive capsulitis and in the shoulders of healthy controls. Please describe better your aim. You can split in a primary and secondary aim."
Response:
We appreciate your detailed suggestions, which have helped us refine the introduction and clarify the study aim. In response to your comments, we have:
- Added a paragraph explaining adhesive capsulitis and its clinical significance, highlighting the importance of ROM assessment in diagnosis and treatment evaluation.
- Incorporated inertial sensors and digital inclinometers as alternative ROM assessment tools, mentioning their advantages and limitations.
- Clarified the study aim by explicitly differentiating the primary aim (evaluating reliability and validity of markerless motion capture) and the secondary aim (observing potential differences between healthy individuals and adhesive capsulitis patients).
Revision in the manuscript:
Original:
Traditionally, ROM has been measured using goniometers; however, this method has notable limitations, such as inter-rater variability, time-consuming procedures, and challenges in stabilizing the trunk and shoulder blades, which can lead to measurement errors. Additionally, shoulder movement involves a complex interaction among multiple joints, including the clavicle, scapula, and humerus, collectively referred to as the scapulohumeral rhythm. Thus, assessing shoulder movement in a two-dimensional manner using a goniometer may not be appropriate [2].…Thus, this study aimed to evaluate whether iBalance demonstrates sufficient reliability and validity for measuring shoulder joint movements in the clinical setting through rigorous and multifaceted assessments.
Updated:
Traditionally, ROM has been measured using goniometers; however, this method has notable limitations, such as inter-rater variability, time-consuming procedures, and challenges in stabilizing the trunk and shoulder blades, which can lead to measurement errors. Additionally, shoulder movement involves a complex interaction among multiple joints, including the clavicle, scapula, and humerus, collectively referred to as the scapulohumeral rhythm. Thus, assessing shoulder movement in a two-dimensional manner using a goniometer may not be appropriate [2]. To address these limitations, alternative ROM assessment tools such as inertial sensors and digital inclinometers have been introduced. Inertial sensors, which integrate accelerometers and gyroscopes, provide a portable and cost-effective solution for motion analysis, particularly in shoulder disorders such as adhesive capsulitis [3]. Similarly, digital inclinometers offer high accuracy in measuring scapular movement and are widely used in rehabilitation settings [4]. However, both methods require direct sensor placement on the body, which can lead to variability in measurements due to sensor positioning and movement artifacts. … Moreover, most motion analysis studies have primarily focused on assessing ROM in healthy individuals, with limited research on patients with shoulder disorders. Adhesive capsulitis, or frozen shoulder, is a common musculoskeletal condition characterized by progressive pain and restricted active and passive ROM without structural lesions [6]. Since passive ROM assessment is crucial for evaluating disease progression and treatment response, expanding motion analysis research to include both active and passive ROM in adhesive capsulitis patients is essential. …Thus, the primary aim of this study was to evaluate the reliability and validity of iBalance for measuring shoulder joint ROM in clinical settings. Additionally, we explored whether the system's measurement reliability and validity showed different trends between healthy individuals and patients with adhesive capsulitis.
- Methods Revision
Reviewer’s Comments:
- "Researchers measured shoulder joint ROM” could you kindly specify if researchers are physiotherapists, engineers, or others?"
Response:
We appreciate your comment. The ROM measurements in this study were conducted by two Korean Medicine doctors with two years of experience in musculoskeletal assessment.
Revision in the manuscript (2.5 Procedures):
Original:
"Two raters with 2 years of experience conducted the study after sufficient practice based on measurement protocol."
Updated:
"Two Korean Medicine doctors with two years of experience in musculoskeletal assessment conducted the study after undergoing standardized training on the measurement protocol."
- "The manuscript does not clearly justify the chosen sample size or discuss how it ensures sufficient power to detect expected differences between groups."
Response:
The primary focus of this study was to assess the reliability and validity of the iBalance system rather than to conduct statistical comparisons between adhesive capsulitis patients and healthy individuals. Therefore, the sample size was determined based on achieving sufficient power for reliability analysis, rather than detecting differences between groups. The inclusion of patients with adhesive capsulitis aimed to explore potential trends in measurement reliability and validity under restricted ROM conditions, rather than to statistically compare the two groups. The 3:1 ratio of healthy individuals (n = 30) to patients with adhesive capsulitis (n = 10) was chosen to ensure the system’s performance could be evaluated across a range of motion conditions while maintaining recruitment feasibility.
Revision in the manuscript (2.8 Sample Size):
Original:
"This study included both healthy individuals (n = 30) and patients with adhesive capsulitis (n = 10), with a 3:1 ratio chosen based on recruitment feasibility to address the lack of data on patient populations in previous studies."
Updated:
"Since the primary aim of this study was to evaluate the reliability and validity of the system rather than to compare groups statistically, the sample size was determined based on achieving sufficient power for reliability analysis. A 3:1 ratio of healthy individuals (n = 30) to patients with adhesive capsulitis (n = 10) was chosen to ensure the inclusion of individuals with restricted ROM, allowing for a broader evaluation of the system's performance across different movement conditions."
- "Inclusion/exclusion criteria should be better explained."
- "Please specify in which phase are the patients with adhesive capsulitis."
Response:
We have expanded the inclusion and exclusion criteria to provide additional details regarding participant selection. Specifically, we clarified the phase of adhesive capsulitis in which patients were enrolled and provided additional exclusion criteria related to neurological and cognitive impairments that could affect ROM assessment. Also, the study included patients with adhesive capsulitis in the freezing or frozen stage. This information has been incorporated into the revised inclusion criteria.
Revision in the manuscript (2.3 Participants):
Original:
"This study included 40 participants aged 19 years or older, consisting of 30 healthy individuals and 10 patients with AC. Healthy individuals were eligible if they had no history of shoulder joint disease or pain. Patients with AC were included if they exhibited passive motion restrictions in two or more directions (flexion < 165°, abduction < 150°, external rotation < 45°) lasting more than 1 month. Patients with other shoulder diseases, such as rotator cuff tears on ultrasound or MRI, were excluded from the AC group. Common exclusion criteria for both groups included difficulty complying with the clinical trial schedule or inability to participate as determined by the researchers.
Updated:
" This study included 40 participants aged 19 years or older, consisting of 30 healthy individuals and 10 patients with adhesive capsulitis. Healthy individuals were eligible if they had no history of shoulder joint disease or pain. Patients with adhesive capsulitis in the freezing or frozen stage were included based on the presence of passive motion restrictions in at least two directions (flexion < 165°, abduction < 150°, external rotation < 45°) lasting more than 1 month [14]. Participants were excluded if they had a history of major trauma involving the shoulder, symptomatic rotator cuff tears confirmed by ultrasound or MRI, inflammatory joint diseases, or any other structural or systemic disorder that could contribute to shoulder pain or restricted ROM. Additionally, individuals from both groups were excluded if they had difficulty complying with the study schedule or if they had severe musculoskeletal pain, neurological disorders affecting motor function, cognitive impairments interfering with test instructions, or other medical conditions that could compromise accurate ROM assessment.”
- "2.4. Measurement Systems: Please add more references that support the use of this Markerless Motion Capture System."
Response:
We appreciate your valuable comment. Because the current study is the first research investigating the iBalance markerless motion capture system, there are no prior publications directly utilizing this specific device. However, we have included references to studies involving a previous-generation markerless depth-sensor-based system, POM-Checker, which employs similar underlying technology and has shown excellent validity and reliability for assessing shoulder movements and posture analysis. These references demonstrate the scientific soundness and clinical applicability of similar markerless depth sensor-based systems, thus indirectly supporting the validity and reliability of the iBalance system used in our study.
Revision in the manuscript (2.4 Measurement Systems):
Original:
"The camera was positioned 95 cm above the ground and aligned perpendicularly, with a distance of 250 cm from the participant to capture their entire body (Figure 2). Measurement accuracy was influenced by environments with poor infrared visibility or by unsuitable clothing, such as black fabrics."
Updated:
" Since this is the first study utilizing the iBalance system, we referenced studies using similar markerless RGB depth sensor-based motion capture systems, which have already shown high validity and reliability in measuring shoulder kinematics [15,16]. The camera was positioned 95 cm above the ground and aligned perpendicularly, with a distance of 250 cm from the participant to capture their entire body (Figure 2). Measurement accuracy was influenced by environments with poor infrared visibility or by unsuitable clothing, such as black fabrics.”
- Result Revision
Reviewer’s Comment:
- "The quality of your tables should be improved, please use only three lines."
Response:
Thank you for your suggestion regarding table formatting. We have revised all tables in the manuscript to follow the recommended three-line format to enhance clarity and readability. We appreciate your feedback, which has helped improve the overall presentation of the data.
- Discussion Revision
Reviewer’s Comments:
- "I would add a discussion concerning the importance to assess the ROM in Adhesive capsulitis."
- "Your discussion would benefit from a comparison with other studies involved in ROM assessment in adhesive capsulitis and healthy controls. You can use the studies suggested in the introduction section."
- "The main limitation of your study is the sample size that does not allow age, gender stratification, nor a comparison between healthy and adhesive capsulitis."
Response:
We have revised the discussion section to better highlight the importance of ROM assessment in adhesive capsulitis, incorporate comparisons with previous studies, and acknowledge the sample size limitation regarding age and gender stratification.
We sincerely appreciate your insightful comments, which have helped us improve the clarity and completeness of our manuscript. We hope that our responses satisfactorily address your feedback and that the revised manuscript meets your expectations.
Revision in the manuscript:
Original:
In patients with adhesive capsulitis, iBalance exhibited good reliability for flexion, abduction, and passive abduction, though overall reliability and validity scores were lower than those for healthy participants. Validity was excellent only for abduction and passive abduction, while other movements demonstrated weaker validity. These findings are consistent with a previous study [14] that reported high variability in ROM measurements among patients with adhesive capsulitis, likely due to differences in disease progression, stages, and joint stiffness. Additionally, pain severity and fluctuation may contribute to this variability, particularly in intra-rater reliability. Clinicians should consider these factors when interpreting iBalance results in pathological populations.
Second, the relatively small sample size, especially for patients with adhesive capsulitis, limits the generalizability of the findings.
Updated:
In patients with adhesive capsulitis, iBalance exhibited good reliability for flexion, abduction, and passive abduction, though overall reliability and validity scores were lower than those for healthy participants. Validity was excellent only for abduction and passive abduction, while other movements demonstrated weaker validity. These findings are consistent with a previous study [14] that reported high variability in ROM measurements among patients with adhesive capsulitis, likely due to differences in disease progression, stages, and joint stiffness. Additionally, pain severity and fluctuation may contribute to this variability, particularly in intra-rater reliability. Clinicians should consider these factors when interpreting iBalance results in pathological populations. Tveita et al. [18] found that while ROM measurements in adhesive capsulitis were re-producible at the group level, individual variability was substantial. This suggests that clinicians should be cautious when assessing ROM in individual patients, as disease progression and pain responses may lead to significant measurement fluctuations. Furthermore, a study utilizing depth sensor-based motion analysis [19] have shown that adhesive capsulitis patients exhibit significantly lower maximum abduction angles and angular velocities compared to both healthy individuals and other shoulder pathology groups. These findings indicate that the observed reductions in ROM and movement speed among adhesive capsulitis patients are primarily driven by pathological changes in joint mechanics rather than inconsistencies in measurement methodology. This highlights the potential of markerless motion capture systems like iBalance to differentiate movement abnormalities across various shoulder disorders.
Second, the relatively small sample size, especially for patients with adhesive capsulitis, limits the generalizability of the findings. While the 3:1 ratio was chosen based on recruitment feasibility, a more balanced sample may provide stronger statistical comparisons. Third, although this study included both healthy individuals and patients with adhesive capsulitis, differences in age and gender were not separately analyzed. Descriptive statistics showed that age differences existed between groups, which may have influenced ROM variability. Future studies should consider age and gender differences to understand their potential impact on ROM measurements better.
We sincerely appreciate your insightful comments, which have helped us improve the clarity and completeness of our manuscript. We hope that our responses satisfactorily address your feedback and that the revised manuscript meets your expectations.
Reviewer 3 Report
Comments and Suggestions for Authors
The paper presents a well-conducted study that evaluates the reliability and validity of the iBalance system for measuring shoulder range of motion (ROM).
The study demonstrates robust metrological evidence for the system's performance, including excellent inter- and intra-rater reliability for key shoulder movements (flexion, abduction, and passive abduction) with Intraclass Correlation Coefficients (ICCs) greater than 0.90. The use of multiple reliability metrics, such as mean absolute deviation (MAD), standard error of the mean (SEM), and minimum detectable change (MDC), further supports the system's precision and consistency.
The authors provide a comprehensive comparison of the iBalance system with a goniometer, showcasing strong validity for the same key movements with ICCs above 0.85.
Moreover, the study addresses potential limitations, such as the variability observed in patients with adhesive capsulitis and the need for further algorithmic refinements. The inclusion of a well-designed protocol with randomized rater order and repeated measurements ensures the reliability of the findings.
Author Response
We sincerely appreciate your positive feedback and thoughtful evaluation of our study. Your recognition of the methodological rigor, reliability metrics, and comprehensive comparisons strengthens our confidence in the study’s contributions. We are grateful for your insights and encouragement, and we hope that our findings will be valuable for future research and clinical applications.
Thank you again for your time and constructive review.
Round 2
Reviewer 2 Report
Comments and Suggestions for Authors
Well done